# Relationship between tobacco use and body mass index- a propensity score matching analysis of an indian National Survey

**Madhur Verma**[1], **Nitin Kapoor**[2,3], **Ajit Kumar Jaiswal**[4], **Prakash Kumar**[1], **Pritam Halder**[5], **Vandana Esht**[6], **Waseem Mumtaz Ahamed**[6], **Omna Singh**[1], **Rakesh Kakkar**[1], **Sanjay Kalra**[7,8], **Sonu Goel**[5,9,10]*

1 Department of Community and Family Medicine, All India Institute of Medical Sciences Bathinda, India, 2 Dept. of Endocrine, Diabetes and Metabolism, Christian Medical College, Vellore,(Tamil Nadu), India, 3 Non-communicable disease unit, The Baker Heart and Diabetes Institute, Melbourne, Victoria, Australia, 4 International Institute for Population Sciences, Mumbai (Maharashtra) India, 5 Department of Community and Family Medicine and School of Public Health, Postgraduate Institute of Medical Education and Research, Chandigarh, 6 Physical Therapy Department, College of Nursing and Health Sciences, Jazan University, Jizan, Saudi Arabia, 7 Department of Endocrinology, Bharti Hospital, Karnal (Haryana), India, 8 University Centre for Research and Development, Chandigarh University, Mohali, India, 9 School of Medicine, Faculty of Education & Health Sciences, University of Limerick, Ireland, 10 Faculty of Human & Health Sciences at Swansea University, United Kingdom

* sonugoel007@yahoo.co.in, Sonu.Goel@ul.ie

## Abstract

### Background

Tobacco and obesity control is among the major health priorities. Previous studies have mixed opinions about their association. The present study was done to investigate the association between Body Mass Index (BMI) and tobacco use (smokers, smokeless tobacco users and dual users) among Indian adults.

### Methods

Secondary analysis of the fifth National Family Health Survey (2019–21) was conducted that included 724,115 women (15–49 years) and 101,839 men (15–54 years). Nutritional status (BMI) was dependent variable. Current tobacco use was primary independent variable. Using sampling weights, bivariate analysis assessed the association, the determinants were explored using the binary logistic multinomial regression. Propensity Score Matching (PSM) was employed using STATA software to control for potential confounding and strengthen causal inference.

### Results

Weighted prevalence of overweight/obesity and underweight was 38.17%, and 18.05%. Underweight prevalence was highest in smokeless tobacco users (17.09%). Overweight/obesity was highest among smokers (41.62%). Compared to non-users,

**Data availability statement:** This study analyses nationally representative survey data that can be accessed using standard protocols from the DHS website (https://dhsprogram.com/data/available-datasets.cfm)

**Funding:** The author(s) received no specific funding for this work.

**Competing interests:** The authors have declared that no competing interests exist.

tobacco users had higher odds of being underweight (AOR: 1.2; 95% CI: 1.2–1.2) and lower odds of being overweight (0.9; 0.9–0.9). PSM confirmed the BMI lower effect of Tobacco (ATT: 0.159), with a non-significant impact on overweight/obesity (ATT: -0.360).

## Conclusions

We present clear evidence that tobacco use, especially smokeless forms, is significantly associated with undernutrition among Indian adults, while its impact on overweight/obesity remains minimal, which otherwise was more common in smokers. The findings clarify the previously mixed evidence and highlight the nutritional impact of tobacco, reinforcing the need for integrated interventions targeting both tobacco cessation and nutritional improvement.

## Introduction

As the prevalence of Non-communicable diseases increases, the role of risk factors aggravating the onset of these diseases is highlighted. Most of these risk factors are modifiable and can be addressed through behavioural modification. Of the various modifiable risk factors enumerated in the literature, tobacco use and high Body Mass Index (BMI) are the most important and have been studied extensively.[1] Though the global burden attributable to tobacco has decreased between 2000 and 2021, it still remains a significant challenge. Smoking alone contributed to a loss of around 156·5 (130·9 to 181·8) million DALY globally, and chewing tobacco was attributed to 1·6 (1·3 to 1·9) million DALY loss.[2] On the other hand, the prevalence of high BMI has gradually increased between 2000 and 2021 at an annual rate of 1·8% (1·6–1·9), with a global loss of 62·0 (32·0 to 90·7) million Disability Adjusted Life Years (DALY) annually.[2] India is home to the highest number of tobacco consumers in the world. The estimates from India's latest National Family Health Survey-5 (NFHS-5) data conducted between 2019–21 depicted a 38% prevalence of tobacco use among men (15–54 years) consume tobacco, which was higher in rural (43%) compared to urban areas (29%).[3] Tobacco use accounts for nearly 1.35 million deaths every year.[4] Unlike many high-income countries where cigarettes dominate, India witnesses a high prevalence of smokeless tobacco products like khaini, gutkha, and betel quid with tobacco.[5] Due to considerable economic and demographic transition, the prevalence of high BMI or overweight and obesity has also started escalating, depicting a combined prevalence of overweight and obesity in males and females per NFHS-5 to be 44.02% and 41.16%, respectively.[6]

Controlling the burden of two is also among the major agenda of Sustainable Development Goals. Evidence shows that risk factors should be addressed concomitantly and not individually.[7] Understanding their interplay can inform more effective prevention and intervention strategies. Firstly, both high BMI and tobacco use contribute independently to an increased risk of morbidity and

mortality.[8] Obesity is associated with metabolic syndrome, hypertension, dyslipidemia, and insulin resistance, which are precursors to chronic illnesses. Simultaneously, tobacco use exacerbates oxidative stress, inflammation, and endothelial dysfunction, promoting atherogenesis and oncogenesis.[9] Secondly, exploring the co-occurrence of high BMI and tobacco use can uncover behavioural and socio-demographic patterns critical for targeted public health interventions. For instance, certain populations may engage in compensatory behaviours where individuals who quit smoking gain weight due to increased appetite or metabolic changes.[10] Conversely, weight concerns might deter smoking cessation, complicating public health efforts to reduce tobacco use. Understanding these dynamics can help tailor cessation programs that address weight management concurrently, enhancing overall health outcomes.[11] Furthermore, the combined impact of high BMI and tobacco use on healthcare systems underscores the need for such studies.[12] Both conditions lead to substantial healthcare costs due to increased hospitalisations, medication use, and long-term care needs. By elucidating their association, health policymakers can allocate resources more efficiently and design integrated interventions that simultaneously address multiple risk factors. So, we can say that tobacco use and high BMI have a synergistic effect which is beyond their individual contributions, and investigating their association becomes pertinent.[13]

However, the available evidence about the direction of the association between the two is still unclear. Studies from India and abroad have consistently shown an inverse relationship between the two.[13,14] Among tobacco users, the prevalence of low BMI is estimated to be higher for bidi-smokers, followed by cigarette smoking, indicating tobacco use is an independent risk factor for low BMI in the Indian population. The negative impact of tobacco use on BMI can be attributed to various mechanisms. Nicotine suppresses appetite by stimulating the hypothalamus to release neurotransmitters like dopamine and serotonin, which reduce hunger.[15] It also reduces the sense of taste and smell and makes food less appealing. It may also increase metabolic rate, which leads to an increase in the body's energy expenditure. Chewing tobacco directly impacts the digestive system and nutrient absorption, lowering BMI. On the other hand, we do have contrasting findings depicting a positive correlation between tobacco use and High BMI. Nicotine initially suppresses appetite, but chronic use, however, leads to metabolic changes and insulin resistance, causing inefficient glucose utilisation and fat storage.[16] Nicotine withdrawal further exacerbates weight gain by increasing appetite and decreasing basal metabolic rate. Additionally, smoking cessation alters gut microbiota, promoting a higher propensity for weight gain.[17] Such varied evidence tested within non-representative samples necessitates an investigation of the relationship between the two risk factors. Within this context, the NFHS-5 collects data regarding tobacco usage and BMI and provides us with an opportunity to study this debated association and provide conclusive evidence. Therefore, the present study was done to investigate the direction of the association between tobacco use and BMI and examine how the association of BMI and tobacco use varies among smokers, smokeless tobacco users and dual users among Indian adults who participated in a national cross-sectional population-based sample.

## Methodology

### Data source

The present study has utilised data from the fifth round of the National Family and Household Survey (NFHS-5), conducted during 2019–2021.[18] NFHS is a nationally representative cross-sectional large-scale household survey conducted in India. This is a multi-round survey of a nationally representative sample throughout India using a multi-stage stratified cluster sampling approach. Data from NFHS provide robust evidence for the effectiveness of already running programmes and help pave the way by identifying newer unmet needs of the population, which further leads to recognising the necessity for specific new programs. Socio-demographic data was collected from the household, men's, and women's questionnaires, while anthropometric measurements, including height, weight and other measurements, were done by rigorously trained staff in the field itself.

## Sample selection

The NFHS sample is self-weighted at the district level. It follows a two-stage sampling approach for selecting villages and Census Enumeration Blocks (CEBs) as primary sampling units in rural and urban areas. The survey followed the probability proportional to size sampling technique to choose villages/CEBs within each rural/urban stratum. In NFHS-5, a total of 636,699 households, consisting of 724,115 women and 101,839 men, were questioned. For the purpose of our analysis, we excluded pregnant women respondents as they use different criteria for defining overweight and obesity and also excluded adults with missing height and weight data. The detailed sample selection process is depicted in Fig 1.

## Study variables

Dependent variable: The presence of underweight, overweight, and obesity were our primary dependent variables estimated using the BMI due to its interpretability and usage as a measure of the degree of adiposity in an individual. The BMI was calculated as the ratio of weight (in Kg) and height squared (in m) and expressed as $kg/m^2$. In the present survey, respondent weight and height were measured using a Seca 874 digital scale and a Seca 213 stadiometer. The survey staff were rigorously trained to measure the anthropometric parameters accurately. The BMI estimates were further categorised as underweight, normal, overweight, and obese, as per the cut-offs for Asian people, i.e., $23–24.99\,kg/m^2$ for overweight and $\geq 25\,kg/m^2$ for obesity. The above-mentioned are better, as Asian people have higher cardiovascular risks at a lower BMI.[19]

Independent variables: Current tobacco use in any form was our primary independent variable. The response was recorded by asking the question-*"Do you currently chew pan masala or tobacco?"*, *"Do you currently smoke cigarettes or*

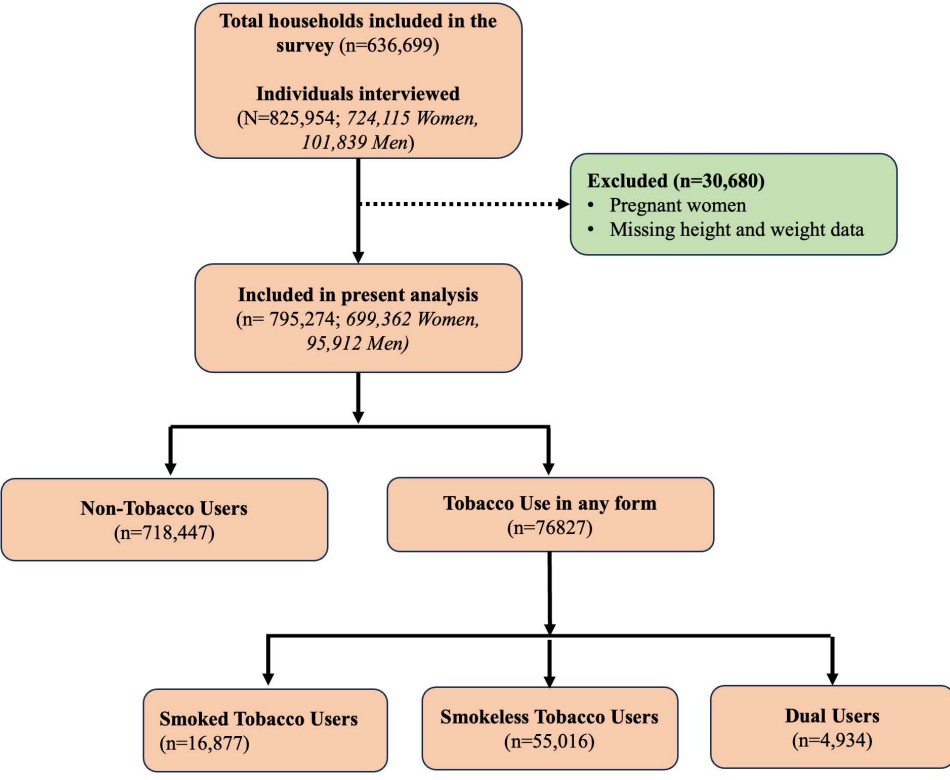

**Fig 1. Flow diagram outlining the sample selection process, particularly in categorising tobacco users by type of tobacco used.**

bidis?", "Do you currently smoke or use tobacco in any other form?" and "In what other forms do you currently smoke or use tobacco?." We categorised individuals as "tobacco users" if the respondent answered "yes" to either one of the first three questions. In addition, we distinguished between tobacco smoking (smoking of cigarettes, bidis, pipes/hookah, and other items) and smokeless tobacco use (consumption of products like ghutka, pan masala, snuff, and khaini), or dual users.

Covariates: The selection of explanatory variables was guided by existing research and the dataset's structure. [10,14,20–22] These included respondent age (categorised as per completed years: 15–24, 25–34, 35–44, and 45–54), gender (men, women), marital status (currently married and not currently married), educational attainment (no education, primary education, and secondary or higher), place of residence (rural and urban), caste groups (categorised as Scheduled Caste/Tribes, other backward classes, and others), religion (categorised as Hindu, Muslim, and Other), region of India (North, Central East, North-East, West, South), socioeconomic status using the wealth quintile. The wealth index is a composite measure of household socioeconomic status. It is typically constructed based on information about household assets, dwelling type, drinking water source, and sanitation facilities in the household. To facilitate analysis, the wealth index is further categorised into five quintiles: poorest (lowest 20%), poorer (lower 20%), middle (middle 20%), wealthier (higher 20%), and richest (highest 20%).

## Statistical analysis

Analysis was done using STATA version 14.2 (Stata Corporation, College Station, Texas, USA). The frequency analysis of the outcome and predictor variables used the sampling weights to ascertain the prevalence (with a 95% confidence interval) of overweight/obesity by different background characteristics. Appropriate weights were used while analysing the prevalence as provided by the NFHS survey. The bivariate analysis was done using the chi-square test of independence to assess the significance of the association between two categorical variables. The factors affecting the likelihood of being overweight/ obese or underweight compared to the Normal BMI in the presence of tobacco use were explored separately using the binary logistic multi-variable regression. We first depicted an unadjusted odds ratio (95% CI) using tobacco(of any type), and then specific type (smoked smokeless vs non-users) variables, followed by an adjusted analysis depicted using adjusted OR (95% CI) was calculated after adjusting for other covariates as well. The p-values <0.05 were considered statistically significant. To assess the specific contribution of tobacco use to BMI outcomes, an additional multivariable regression model without tobacco use is also conducted. Additionally, to control for potential confounding and strengthen causal inference, Propensity Score Matching (PSM) was employed to estimate the effect of tobacco consumption on nutritional outcomes. A one-to-one nearest neighbour matching algorithm without replacement was used, with a calliper and common support applied. Covariates included in the propensity score model were age group, education level, place of residence, and wealth quintile. Post-matching, treatment effects were estimated in terms of ATT (Average Treatment effect on the Treated), ATU (Average Treatment effect on the Untreated), and ATE (Average Treatment Effect). Further, an independent samples t-test was conducted to compare the mean BMI between tobacco users and non-users. This test was used to assess whether the two groups had a statistically significant difference in mean BMI.

## Ethics approval

This study uses publicly available, anonymised data from NFHS-5 (2019–2020), accessed via standard protocols (https://dhsprogram.com/data/available-datasets.cfm). No ethical approval was required due to the anonymised nature of the data.

## Results

Table 1 summarises the sample of 825,954 adults per their socio-demographic and health-related characteristics. The age distribution shows that 33.24%were 15–24 years old, while 29.71%, 24.46%, and 12.58% were in the 25–34, 35–44, and 45–54 years age groups. A higher percentage was female (87.8%), and 12.2% were male. Urban residents constitute

**Table 1.  Sample characteristics of the participants included in the fifth round of the National Family Health Survey India (2019–21).**

| Variables | Unweighted counts | Weighted Percent |
|---|---|---|
| **Total** | **825,954** | **100** |
| **Age (Completed years)** | | |
| 15-24 | 273,415 | 33.24 |
| 25-34 | 247,197 | 29.71 |
| 35-44 | 202,812 | 24.46 |
| 45-54 | 102,530 | 12.58 |
| **Gender** | | |
| Male | 101,839 | 12.2 |
| Female | 724,115 | 87.8 |
| **Place of Residence** | | |
| Urban | 205,955 | 32.21 |
| Rural | 616,999 | 67.8 |
| **Region** | | |
| North | 168,749 | 14 |
| Central | 193,244 | 24 |
| East | 133,554 | 22.78 |
| North-East | 118,293 | 3.75 |
| West | 83,429 | 14.28 |
| South | 128,685 | 20.59 |
| **Highest Level of Education** | | |
| No Education | 169,788 | 20.07 |
| Primary | 102,820 | 12.51 |
| Secondary | 434,218 | 51.55 |
| Higher | 118,907 | 15.86 |
| **Marital status** | | |
| Never Married | 217,698 | 25.19 |
| Currently Married | 577,621 | 71.07 |
| Widowed/Divorced/Separated | 30,635 | 3.74 |
| **Religion** | | |
| Hindu | 623,218 | 81.44 |
| Muslim | 102,841 | 13.42 |
| Other | 99,895 | 5.14 |
| **Social Caste** | | |
| Scheduled Caste | 159,197 | 21.86 |
| Scheduled Tribe | 154,593 | 9.35 |
| Other Backward Caste | 316,207 | 43.01 |
| Other | 195,957 | 25.78 |
| **Wealth Quintile** | | |
| Poorest | 169,640 | 18.44 |
| Poorer | 182,939 | 20.11 |
| Middle | 173,220 | 20.64 |
| Richer | 159,816 | 20.84 |
| Richest | 140,339 | 19.97 |
| **Body Mass Index (BMI)[*]** | | |
| Underweight | 138,897 | 18.05 |
| Normal | 361,104 | 43.78 |

*(Continued)*

**Table 1.** (Continued)

| Variables | Unweighted counts | Weighted Percent |
|---|---|---|
| Overweight/Obese | 295,273 | 38.17 |
| **Tobacco use** | | |
| Non-users | 749,127 | 93.04 |
| Smoked Tobacco Users | 16,877 | 1.73 |
| Smokeless Tobacco Users | 55,016 | 4.78 |
| Dual Users | 4,934 | 0.46 |

*for only 795,274 participants, BMI information was available.

32.21% of the sample. About 18.05% of the participants had a BMI within a normal range, 43.78% were underweight, and 38.17% were either overweight or obese. Most individuals (93.04%) were non-tobacco users, while 1.73% were smoking, 4.78% had smokeless tobacco, and 0.46% were dual users.

We subsequently distributed the participants according to their BMI status and segregated them as per their socio-demographic characteristics (Table 2). Overall, we observed the weighted prevalence of overweight/obesity to be around 38.17%, while underweight was 18.05%. Age shows a contrasting trend where the proportion of underweight individuals decreases, and obesity increases with age, with the oldest age group (45–54) having the highest obesity at 54.6%. Gender differences indicate that males are slightly more overweight/obese (40.6%) than females (37.8%), while under-weight females depicted a higher proportion than males. Likewise, the overweight/obese prevalence was higher in Urban residents (48.8%), those having higher education levels (45.1%), widowed/divorced/separated participants (45.7%), religion other than Hinduism or Muslim Religion (46.3%), Others' social caste (44.8%), higher wealth quintile (54.9%) and participants from South India (50.5%). Contrary to this, the proportion of underweight was highest in rural (20.4%), with fewer years of education. (20.3%), currently married (13.1%), following Hinduism (18.7%), scheduled tribe (24.2%), and poorest wealth quintile (26.9%), and from the Western Region of India (21.4%)

Overall, the mean BMI for tobacco non-users was significantly higher (22.31 kg/m$^2$±0.43), compared to tobacco users (22.17 kg/m$^2$±0.39). Following this, we further segregated the BMI classification as per the Tobacco usage (Table 3). It was seen that the prevalence of underweight was highest in smokeless tobacco users (17.09%), while overweight/obesity was highest among smoked tobacco users (41.62%). Dual users depicted the lowest prevalence of underweight compared to the other two groups. The distribution of smoked and smokeless tobacco users and dual usage depicted significant disparities across all socio-demographic variables.

The initial multinomial binary logistic regression analysis was done after considering tobacco in any form as an independent variable (S1 Table). Both unadjusted and adjusted binary logistic regression analyses reveal a significant association between tobacco use and nutritional status. In the unadjusted model, tobacco users had 15% and 8% lower odds of being underweight (OR: 0.85; 95% CI: 0.83–0.87) and overweight (0.92; 0.91–0.90) than non-users. However, after adjusting for sociodemographic factors, the association reversed: tobacco users had 20% higher odds of being underweight (AOR: 1.2; 95% CI: 1.2–1.2) and 10% lower odds of being overweight (0.9; 0.9–0.9). Following this, Table 4 further explores the odds of having altered BMI due to tobacco as per the type of tobacco. Adjusted analysis depicted that tobacco significantly affected the BMI, and participants consuming smoked and smokeless forms of tobacco depicted a higher likelihood of being underweight (aOR = 1.12; 95% CI: 1.18–1.07 and 1.25;1.29–1.22), and a lesser likelihood of being overweight/obese (0.92; 0.95–0.89 and 0.87; 0.89–0.85) than being in normal BMI category, when compared to a non-tobacco user. Smokeless tobacco users depicted higher chances of being underweight than smokers, while the latter depicted higher chances of being overweight/obese. The likelihood of being underweight decreased with age, education,

**Table 2. Distribution of adults who participated in the fifth round of the National Family Health Survey (2019–21) as per their BMI category.**

| Variables | Total | Underweight | | Normal | | Overweight/Obese | | p-value |
|---|---|---|---|---|---|---|---|---|
| | | Unweighted Counts | Weighted % | Unweighted Counts | Weighted % | Unweighted Counts | Weighted % | |
| **Total** | 795,274 | 138,897 | 18.1 | 361,104 | 43.8 | 295,274 | 38.2 | |
| **Age** | | | | | | | | <0.001 |
| 15-24 | 261,323 | 78435 | 31.3 | 137673 | 50.7 | 137673 | 18.0 | |
| 25-34 | 238,649 | 32253 | 13.9 | 109437 | 44.2 | 109437 | 41.9 | |
| 35-44 | 196,151 | 19097 | 9.8 | 76832 | 37.9 | 76832 | 52.4 | |
| 45-54 | 99,151 | 9112 | 9.2 | 37162 | 36.2 | 37162 | 54.6 | |
| **Gender** | | | | | | | | <0.001 |
| Male | 95,912 | 13908 | 15.5 | 42896 | 44.0 | 42896 | 40.6 | |
| Female | 699,362 | 124989 | 18.4 | 318208 | 43.8 | 318208 | 37.8 | |
| **Place of Residence** | | | | | | | | <0.001 |
| Urban | 194,454 | 24433 | 13.0 | 75587 | 38.2 | 75587 | 48.8 | |
| Rural | 600,820 | 114464 | 20.4 | 285517 | 46.3 | 285517 | 33.3 | |
| **Highest Education** | | | | | | | | <0.001 |
| No Education | 164,453 | 28092 | 17.4 | 78371 | 46.8 | 78371 | 35.8 | |
| Primary | 99,700 | 15754 | 15.9 | 45787 | 44.2 | 45787 | 39.9 | |
| Secondary | 418,316 | 80626 | 20.3 | 188314 | 43.1 | 188314 | 36.6 | |
| Higher | 112,595 | 14393 | 13.3 | 48530 | 41.7 | 48530 | 45.1 | |
| **Marital status** | | | | | | | | <0.001 |
| Never Married | 206,412 | 62621 | 33.0 | 106498 | 49.4 | 106498 | 17.6 | |
| Currently Married | 559,200 | 72303 | 13.1 | 241837 | 42.0 | 241837 | 44.9 | |
| Widowed/Divorced/Separated | 12,920 | 3973 | 13.2 | 12769 | 41.1 | 12769 | 45.7 | |
| **Religion** | | | | | | | | <0.001 |
| Hindu | 600,803 | 115044 | 18.7 | 271697 | 44.0 | 271697 | 37.3 | |
| Muslim | 97,498 | 13795 | 15.7 | 43587 | 43.6 | 43587 | 40.7 | |
| Other | 96,973 | 10058 | 13.9 | 45820 | 39.8 | 45820 | 46.3 | |
| **Social Caste** | | | | | | | | <0.001 |
| Scheduled Caste | 153,146 | 30514 | 19.7 | 69671 | 45.1 | 69671 | 35.2 | |
| Scheduled Tribe | 151,078 | 26788 | 24.2 | 78657 | 50.9 | 78657 | 24.9 | |
| Other Backward Caste | 303,865 | 56716 | 18.1 | 135131 | 43.2 | 135131 | 38.6 | |
| Others | 187,185 | 24879 | 14.2 | 77645 | 40.9 | 77645 | 44.8 | |
| **Wealth Quintile** | | | | | | | | <0.001 |
| Poorest | 164,639 | 41680 | 26.9 | 88355 | 52.1 | 88355 | 21.0 | |
| Poorer | 177,637 | 35781 | 21.9 | 88414 | 48.6 | 88414 | 29.5 | |
| Middle | 167,734 | 27904 | 17.7 | 75679 | 43.9 | 75679 | 38.4 | |
| Richer | 153,690 | 20737 | 14.1 | 62246 | 39.5 | 62246 | 46.4 | |
| Richest | 131,574 | 12795 | 10.0 | 46410 | 35.1 | 46410 | 54.9 | |
| **Region of India** | | | | | | | | <0.001 |
| North | 160,932 | 21634 | 14.8 | 69171 | 44.2 | 69171 | 40.9 | |
| Central | 184,124 | 38024 | 19.6 | 89842 | 47.5 | 89842 | 32.8 | |
| East | 129,599 | 28984 | 20.1 | 62482 | 47.3 | 62482 | 32.7 | |
| North-East | 116,264 | 12766 | 15.0 | 59862 | 51.5 | 59862 | 33.5 | |
| West | 80,810 | 18447 | 21.4 | 34623 | 41.6 | 34623 | 37.0 | |
| South | 123,545 | 19042 | 14.4 | 45124 | 35.2 | 45124 | 50.5 | |

**Table 3. Distribution of NFHS-5 adult study participants as per their BMI, Tobacco type usage, segregated as per different socio-demographic characteristics.**

| Variables | Smoked tobacco users | | | | | Smokeless tobacco users | | | | | Dual tobacco users | | | | |
|---|---|---|---|---|---|---|---|---|---|---|---|---|---|---|---|
| | Total | Under weight | Normal | Over-weight/ Obese | p-value | Total | Under weight | Normal | Over-weight/ Obese | p-value | Total | Under weight | Normal | Over-weight/ Obese | p-value |
| **Total sample size** | 16,115 | 2,135 | 7,268 | 6712 | | 53,562 | 9,154 | 25,880 | 18,528 | | 4,708 | 502 | 2,314 | 1,892 | |
| **Overall (weighted %)** | | 13.25 | 45.1 | 41.62 | | | 17.09 | 48.32 | 34.59 | | | 10.66 | 49.15 | 40.19 | |
| **Age** *(completed years)* | | | | | <0.001 | | | | | <0.001 | | | | | <0.001 |
| 15-24 | 2,864 | 26.6 | 53.1 | 20.4 | | 7,828 | 31.7 | 54.0 | 14.4 | | 1020 | 24.4 | 54.0 | 21.6 | |
| 25-34 | 4,785 | 15.3 | 44.7 | 40.1 | | 16,716 | 20.0 | 48.8 | 31.2 | | 1621 | 11.6 | 51.4 | 37.0 | |
| 35-44 | 4,830 | 12.8 | 38.7 | 48.5 | | 18,490 | 17.1 | 44.9 | 38.0 | | 1299 | 7.3 | 45.6 | 47.1 | |
| 45-54 | 3,636 | 11.4 | 41.7 | 46.9 | | 10,528 | 15.0 | 43.5 | 41.6 | | 768 | 9.5 | 42.5 | 48.0 | |
| **Sex** | | | | | <0.001 | | | | | <0.001 | | | | | 0.859 |
| Male | 11249 | 12.2 | 43.1 | 44.7 | | 13969 | 14.8 | 48.6 | 36.5 | | 3890 | 12.7 | 49.5 | 37.8 | |
| Female | 4866 | 24.6 | 45.2 | 30.2 | | 29593 | 22.0 | 46.1 | 31.9 | | 818 | 18.5 | 46.0 | 35.5 | |
| **Place of Residence** | | | | | <0.001 | | | | | <0.001 | | | | | <0.001 |
| Urban | 3741 | 11.2 | 37.4 | 51.4 | | 9828 | 13.5 | 40.3 | 46.3 | | 1157 | 10.9 | 45.1 | 44.0 | |
| Rural | 12374 | 17.8 | 46.7 | 35.5 | | 43734 | 21.1 | 48.8 | 30.1 | | 3551 | 14.3 | 51.0 | 34.7 | |
| **Region of India** | | | | | <0.001 | | | | | <0.001 | | | | | <0.001 |
| North | 3,425 | 11.0 | 40.0 | 49.0 | | 3,424 | 18.3 | 49.3 | 32.4 | | 451 | 10.6 | 54.5 | 34.9 | |
| Central | 2,608 | 18.4 | 46.4 | 35.3 | | 12,784 | 20.1 | 48.8 | 31.1 | | 1,227 | 12.4 | 52.4 | 35.2 | |
| East | 2,285 | 16.6 | 48.4 | 35.0 | | 6,775 | 21.9 | 47.6 | 30.5 | | 756 | 16.6 | 47.5 | 35.9 | |
| Northeast | 4,760 | 10.8 | 50.9 | 38.3 | | 20,888 | 13.6 | 50.2 | 36.3 | | 1,794 | 9.1 | 51.4 | 39.5 | |
| West | 952 | 21.7 | 38.8 | 39.6 | | 6,506 | 21.1 | 42.9 | 36.0 | | 271 | 10.5 | 45.1 | 44.4 | |
| South | 2,085 | 13.5 | 38.6 | 48.0 | | 3,185 | 17.0 | 40.0 | 43.1 | | 209 | 16.4 | 35.9 | 47.7 | |
| **Highest Education** | | | | | <0.001 | | | | | <0.001 | | | | | <0.001 |
| No Education | 3,456 | 21.2 | 46.8 | 32.0 | | 17,074 | 23.2 | 47.8 | 29.0 | | 789 | 17.1 | 54.0 | 28.9 | |
| Primary | 2,456 | 16.4 | 48.5 | 35.1 | | 10,620 | 19.6 | 47.5 | 32.9 | | 796 | 14.8 | 51.2 | 34.0 | |
| Secondary | 8,303 | 14.9 | 42.0 | 43.2 | | 23,152 | 17.0 | 46.4 | 36.5 | | 2,623 | 12.2 | 48.5 | 39.3 | |
| Higher | 1,892 | 8.6 | 39.1 | 52.3 | | 2,683 | 9.7 | 42.4 | 47.9 | | 497 | 7.7 | 40.9 | 51.4 | |
| **Marital status** | | | | | <0.001 | 33 | | | | <0.001 | 3 | | | | <0.001 |
| Never Married | 3,454 | 21.6 | 50.2 | 28.2 | | 6,838 | 28.1 | 52.0 | 19.9 | | 1,263 | 19.4 | 51.9 | 28.6 | |
| Currently Married | 12,023 | 13.7 | 41.7 | 44.6 | | 42,924 | 18.3 | 46.3 | 35.5 | | 3,276 | 10.7 | 48.5 | 40.8 | |

*(Continued)*

**Table 3.** (Continued)

| Variables | Smoked tobacco users | | | | | Smokeless tobacco users | | | | | Dual tobacco users | | | | |
| | Total | Under weight | Normal | Over-weight/ Obese | p-value | Total | Under-weight | Normal | Over-weight/ Obese | p-value | Total | Under-weight | Normal | Over-weight/ Obese | p-value |
|---|---|---|---|---|---|---|---|---|---|---|---|---|---|---|---|
| Widowed/ Divorced/ Separated | 638 | 20.5 | 46.6 | 32.8 | | 3,800 | 20.4 | 47.1 | 32.5 | | 169 | 13.6 | 39.3 | 47.1 | |
| **Religion** | | | | | <0.001 | | | | | <0.001 | | | | | <0.001 |
| Hindu | 10,395 | 16.6 | 43.1 | 40.3 | | 35,366 | 20.7 | 46.9 | 32.4 | | 3,038 | 13.6 | 49.3 | 37.0 | |
| Muslim | 2,067 | 10.8 | 46.2 | 42.9 | | 5,427 | 14.6 | 46.3 | 39.1 | | 339 | 11.8 | 51.4 | 36.9 | |
| Other | 3,653 | 13.2 | 45.9 | 40.8 | | 12,769 | 13.1 | 50.5 | 36.5 | | 1,331 | 10.7 | 41.2 | 48.1 | |
| **Social Caste** | | | | | <0.001 | | | | | <0.001 | | | | | <0.001 |
| Scheduled Caste | 2,696 | 18.6 | 42.1 | 39.4 | | 8,990 | 19.9 | 47.3 | 32.8 | | 841 | 15.7 | 50.1 | 34.2 | |
| Scheduled Tribe | 5,480 | 24.2 | 49.0 | 26.8 | | 21,007 | 24.6 | 51.8 | 23.6 | | 1,856 | 13.3 | 53.8 | 32.9 | |
| Other Backward Caste | 4,334 | 14.1 | 42.7 | 43.2 | | 14,579 | 19.0 | 45.7 | 35.3 | | 1,317 | 12.6 | 48.3 | 39.2 | |
| Others | 3,605 | 10.5 | 43.5 | 46.0 | | 8,986 | 14.6 | 43.9 | 41.6 | | 694 | 11.2 | 45.9 | 42.9 | |
| **Wealth Quintile** | | | | | <0.001 | | | | | <0.001 | | | | | <0.001 |
| Poorest | 4,016 | 25.6 | 51.2 | 23.2 | | 19,231 | 26.9 | 52.0 | 21.2 | | 1,328 | 19.6 | 53.1 | 27.4 | |
| Poorer | 3,955 | 19.0 | 48.3 | 32.6 | | 15,092 | 19.5 | 49.3 | 31.2 | | 1,317 | 13.7 | 53.1 | 33.2 | |
| Middle | 3,409 | 13.9 | 44.6 | 41.5 | | 10,323 | 15.2 | 43.9 | 40.9 | | 973 | 9.9 | 50.8 | 39.2 | |
| Richer | 2,766 | 9.3 | 38.1 | 52.7 | | 6,318 | 11.1 | 39.3 | 49.6 | | 706 | 11.2 | 40.5 | 48.3 | |
| Richest | 1,969 | 7.7 | 33.1 | 59.2 | | 2,598 | 7.1 | 33.5 | 59.5 | | 384 | 5.2 | 38.3 | 56.5 | |

and wealth status. But it was higher for females, who were residents of rural South India. Likewise, the odds of being over-weight or obese increased with age, male gender, urban areas, South India, more years of education, and wealth status.

Table 5 presents the propensity score matching (PSM) estimates evaluating the impact of tobacco consumption on nutritional outcomes. In the unmatched sample, 15.9% of tobacco users were underweight compared to 17.6% of non-users (difference: -0.018; t = -12.18; p < 0.001). After matching, the average treatment effect on the treated (ATT) was 0.159 (t = 118.37), indicating a 15.9 percentage point higher likelihood of being underweight among tobacco users. The average treatment effect on the untreated (ATU) was -0.176, and the average treatment effect (ATE) was -0.145, with a standard error of 0.001. For overweight status, unmatched prevalence was 36.5% among users and 37.2% among non-users (difference: -0.007; t = -3.87; p < 0.001). Following PSM, ATT was -0.360 (t = -1.72), ATU was 0.165, and ATE was 0.116, with a standard error of 0.209. The ATT and ATE estimates for overweight were not statistically significant. The sample included 74,385 treated individuals and 720,889 untreated individuals.

## Discussion

With technological advancements in the modern lifestyle, the increasing prevalence of unhealthy behaviour, like inappropriate dietary habits and substance abuse, has been documented over the years.[23] These risk factors continue to disrupt the metabolic profile as they work synergistically when together but lack robust scientific evidence. Global studies depicting the effect of tobacco on BMI have been controversial and non-confirmatory. The present study is the first from India to provide valuable

**Table 4. Unadjusted and adjusted odds of being underweight or overweight/obese due to tobacco use as per the Multinomial binary logistic regression.**

| Covariates | Underweight | | | | Overweight/ Obese | | | |
|---|---|---|---|---|---|---|---|---|
| | Unadjusted Odds ratio (95% CI) | p-values | Adjusted Odds ratio (95% CI) | p-values | Unadjusted Odds ratio (95% CI) | p-values | Adjusted Odds ratio (95% CI) | p-values |
| **Tobacco usage** | | | | | | | | |
| None ® | Reference | | Reference | | Reference | | Reference | |
| Smoke tobacco users | 0.75(0.79-0.72) | <0.001 | 1.12(1.18-1.07) | <0.001 | 1.12(1.16-1.08) | 0.001 | 0.92(0.95-0.89) | <0.001 |
| Smokeless tobacco users | 0.91(0.93-0.88) | <0.001 | 1.25(1.29-1.22) | <0.001 | 0.87(0.89-0.85) | 0.001 | 0.87(0.89-0.85) | <0.001 |
| Dual Users | 0.56(0.61-0.50) | <0.001 | 0.81(0.89-0.73) | <0.001 | 0.99(1.06-0.93) | 0.821 | 0.98(1.05-0.92) | 0.516 |
| **Age (Completed years)** | | | | | | | | |
| 15-24® | | | Reference | | | | Reference | |
| 25-34 | | | 0.65(0.66-0.64) | <0.001 | | | 2.14(2.18-2.1) | <0.001 |
| 35-44 | | | 0.52(0.53-0.5) | <0.001 | | | 3.28(3.34-3.22) | <0.001 |
| 45-54 | | | 0.5(0.52-0.49) | <0.001 | | | 3.64(3.72-3.56) | <0.001 |
| **Sex** | | | | | | | | |
| Male® | | | Reference | | | | Reference | |
| Female | | | 1.32(1.34-1.29) | <0.001 | | | 0.88(0.89-0.86) | <0.001 |
| **Place of Residence** | | | | | | | | |
| Urban® | | | Reference | | | | Reference | |
| Rural | | | 1.07(1.09-1.05) | <0.001 | | | 0.86(0.88-0.85) | <0.001 |
| **Region of India** | | | | | | | | |
| North® | | | Reference | | | | Reference | |
| Central | | | 1.11(1.14-1.09) | <0.001 | | | 0.88(0.9-0.87) | 0.001 |
| East | | | 1.23(1.26-1.2) | <0.001 | | | 0.94(0.95-0.92) | 0.001 |
| Northeast | | | 0.67(0.69-0.66) | <0.001 | | | 1.04(1.06-1.02) | 0.001 |
| West | | | 1.75(1.79-1.7) | <0.001 | | | 0.8(0.82-0.78) | 0.001 |
| South | | | 1.39(1.42-1.36) | <0.001 | | | 1.34(1.36-1.31) | 0.001 |
| **Highest Education** | | | | | | | | |
| No Education® | | | Reference | | | | Reference | |
| Primary | | | 0.91(0.93-0.89) | <0.001 | | | 1.14(1.16-1.12) | <0.001 |
| Secondary | | | 0.87(0.88-0.85) | <0.001 | | | 1.28(1.3-1.26) | <0.001 |
| Higher | | | 0.61(0.63-0.6) | <0.001 | | | 1.39(1.42-1.37) | <0.001 |
| **Marital status** | | | | | | | | |
| Never Married® | | | Reference | | | | Reference | |
| Currently Married | | | 0.6(0.61-0.59) | <0.001 | | | 1.72(1.76-1.69) | <0.001 |
| Widowed/Divorced/Separated | | | 0.66(0.69-0.63) | <0.001 | | | 1.55(1.59-1.5) | <0.001 |
| **Religion** | | | | | | | | |
| Hindu® | | | Reference | | | | Reference | |
| Muslim | | | 0.76(0.78-0.74) | <0.001 | | | 1.21(1.23-1.19) | <0.001 |
| Other | | | 0.72(0.74-0.7) | <0.001 | | | 1.28(1.3-1.25) | <0.001 |
| **Social Caste** | | | | | | | | |
| Scheduled Caste® | | | Reference | | | | Reference | |
| Scheduled Tribe | | | 0.89(0.91-0.87) | <0.001 | | | 0.84(0.85-0.82) | <0.001 |
| Other Backward Caste | | | 0.99(1-0.97) | 0.13 | | | 0.97(0.98-0.95) | <0.001 |
| Others | | | 0.91(0.93-0.89) | <0.001 | | | 1.14(1.16-1.12) | <0.001 |
| **Wealth Quintile** | | | | | | | | |
| Poorest® | | | Reference | | | | Reference | |

*(Continued)*

**Table 4.** (Continued)

| Covariates | Underweight | | | | Overweight/ Obese | | | |
|---|---|---|---|---|---|---|---|---|
| | Unadjusted Odds ratio (95% CI) | p-values | Adjusted Odds ratio (95% CI) | p-values | Unadjusted Odds ratio (95% CI) | p-values | Adjusted Odds ratio (95% CI) | p-values |
| Poorer | | | 0.84(0.86-0.83) | <0.001 | | | 1.45(1.47-1.42) | <0.001 |
| Middle | | | 0.74(0.75-0.72) | <0.001 | | | 1.89(1.92-1.86) | <0.001 |
| Richer | | | 0.67(0.69-0.65) | <0.001 | | | 2.39(2.44-2.35) | <0.001 |
| Richest | | | 0.6(0.61-0.58) | <0.001 | | | 3.04(3.12-2.98) | <0.001 |
| **Constant** | | <0.001 | 0.74(0.77-0.71) | <0.001 | | <0.001 | 0.14(0.14-0.13) | <0.001 |

**Table 5. Propensity Score Matching Estimates of the Effect of Tobacco Use on Underweight and Overweight Status.**

| Nutritional Outcome | Estimate Type | Treated | Controls | Difference | S.E. | T-stat | ATE | S.E. (ATE) |
|---|---|---|---|---|---|---|---|---|
| **Sample Size (Tobacco users/ Non-users)** | | | | | | | | |
| **Underweight vs Non-Underweight** | Unmatched | 0.159 | 0.176 | -0.018 | 0.001 | -12.18 | | |
| 74,385/720,889 | ATT | 0.159 | 0 | | 0.001 | 118.37 | | |
| | ATU | 0.176 | 0 | | | | | |
| | ATE | | | | | | -0.145 | 0.001 |
| **Overweight vs Non-Overweight** | Unmatched | 0.365 | 0.372 | -0.007 | 0.002 | -3.87 | | |
| 74,385/720,889 | ATT | 0.365 | 0.724 | -0.36 | 0.209 | -1.72 | | |
| | ATU | 0.372 | 0.537 | 0.165 | | | | |
| | ATE | | | | | | 0.116 | 0.209 |

*ATT: Average treatment effect on the treated (tobacco use). ATU: Average treatment effect on the untreated (non-users). ATE: Average treatment effect*

insights using a nationally representative dataset. We observed certain interesting findings. First, the prevalence of overweight and obesity is high among Indian adults. Second, the proportion of underweight adults was highest in the smokeless tobacco users, but overweight and obese were highest in the smoker's group. Third, there were higher odds of being overweight/obese or malnourished in the smokeless tobacco users, with higher odds for the latter, while the protective effect of smoking tobacco towards overweight/obese was lesser compared to being underweight. Lastly, the results from PSM suggest a higher probability of being underweight than comparable non-users, with a robust negative nutritional impact of tobacco use on body weight.

Our study confirms the positive association between tobacco use and under-malnutrition, irrespective of the smoked or smokeless forms and BMI and the odds were higher for smokeless tobacco consumers. Among the 42 populations evaluated for the WHO Multinational Monitoring of Trends and Determinants in Cardiovascular Disease (MONICA) project, it emerged that frequent smokers had considerably lower BMI in 20 populations for men and 30 populations for women.[24] A community-level study from Delhi, the national capital of India, also observed higher odds of being underweight among smokers, but the study did not explore the association with smokeless tobacco users.[14] Higher risk was documented among men (80%) than women (60%) in Mumbai.[13] The relationship between smoking and underweight was not as strong among males in groups where the percentage of regular smokers was higher than the percentage of ex-smokers. Another historical study from the United States of America depicted the weight loss effect of tobacco smoking.[25] A study from the United Kingdom population also corroborated the negative association between smoking and BMI.[26] The association between tobacco use (both smoked and smokeless) and BMI involves complex physiological mechanisms and socio-demographic factors. Although the precise causes of these relationships remain unclear, several theories include the energy Expenditure theory (Smokers' lower BMI may be attributed to their higher overall energy expenditure) and the

Plasma Leptin theory (this hormone regulates energy intake and expenditure).[27] Smoked tobacco, primarily through nicotine, suppresses appetite and increases metabolic rate, often leading to lower BMI.[28] Physiologically, both forms of tobacco alter energy homeostasis through effects on leptin and ghrelin levels.[29] However, smokeless tobacco users may not experience the same appetite suppression as smokers.[30]

We observed that the smoker group depicted the highest proportion of overweight and obese adults. Some studies have depicted an increase in weight in smokers, but in special contexts. In a study among Swiss adolescents, heavy smokers were more likely to be overweight than non-smokers.[31] Another study among Japanese workers found that current smokers had higher BMI compared to non-smokers, particularly among those with low physical activity levels. [32] Compared to never-smokers, male smokers in the FINRISK investigations had a higher likelihood of being obese. [33] In the health survey in Switzerland, the adjusted risks of obesity were lower among non-smokers and light smokers and greater among previous smokers and heavy smokers.[31] Previous reviews explain these findings in different ways. While the current secondary analysis restrains us from looking into tobacco consumption habits more comprehensively, it is reported that heavy smokers tend to have greater body weight than light smokers or non-smokers, which likely reflects a clustering of risky behaviours like inadequate physical activity, increased sedentariness, inappropriate diet, and concurrent substance abuse like alcohol that is conducive to weight gain.[34] In addition, smoking increases insulin resistance and is associated with central fat accumulation. Compared to non-smokers, current smokers frequently have a greater waist circumference (WC) and waist-to-hip ratio (WHR).[35] A recent study reported a positive association of smoking with multimorbidity, which is a determinant of higher BMI.[36] As a result, smoking increases the risk of metabolic syndrome and diabetes, and these factors increase the risk of cardiovascular disease.[37]

We observed lower odds of being obese/overweight among smokers compared to non-smokers and were even lower for smokeless tobacco users. The differences in the pharmacokinetics of smoked and smokeless tobacco can explain the differences.[38] While a study that primarily focuses on water pipes and cigarettes, provides insights into how different forms of tobacco use, including smokeless tobacco, influence nicotine exposure over time.[39] Smokeless tobacco typically results in higher cumulative levels of nicotine in the blood that are more sustained compared to smoked tobacco. [40] This is due to the way nicotine is absorbed and metabolised with each form of tobacco use. When smoking, nicotine is rapidly absorbed through the lungs, leading to a sharp spike in blood nicotine levels. However, these levels decline relatively quickly as the nicotine is metabolised. Smoking generally produces more frequent but shorter nicotine spikes. In contrast, nicotine from smokeless tobacco is absorbed more slowly through the mucous membranes (oral or nasal). However, the absorption continues over a longer period, resulting in more sustained and elevated nicotine levels in the bloodstream. The cumulative exposure (total amount of nicotine in the blood over time) is often higher than that from smoking. A study comparing cigarette smokers and smokeless tobacco users found that those using smokeless products had higher 24-hour nicotine exposure despite using fewer doses of the product compared to the number of cigarettes smoked.[41]

We observed certain socio-demographic **v**ariables emerged as significant predictors of BMI outcomes in the presence of tobacco use, depicting the disparities. The inconsistent evidence about the direction and strength of the relationship between smoking and BMI is affected by various socio-cultural, behavioural and demographic characteristics of the population.[20,42] Socio-demographically, tobacco use generally correlates with lower socioeconomic status, which is independently associated with higher BMI.[43,44] However, many potential confounders like education level, dietary habits, and physical activity need to be addressed.[26] The impact on BMI varies across populations, with stronger associations often observed in younger age groups and women [7]. Cultural factors also influence the prevalence of smokeless tobacco use in certain communities [8].

There are certain strengths and limitations of the study that should be acknowledged. While most of the previous studies testing the association between tobacco and BMI are limited by the sample size, the present study stands out as one of the largest to date from India. The data has been collected using robust sampling methodology and standardised measurements that make the estimates reliable. Anthropometric measurements were done per international standards

and following rigorous training that allowed international comparison. The weighted analysis of the nationally representative data makes the results generalisable. However, the biggest limitation is the cross-sectional nature of the survey, which restrains us from establishing the temporality between the tobacco and BMI changes over time limitations. The questionnaire's self-reporting style has a high chance of recall bias, misclassification bias, and social desirability bias, which might contribute to deviation from the actual pragmatic scenario. Secondary data analysis is also inhibited by the pre-existing number of variables that can be used for the analysis. For instance, we could not measure the lifetime consumption of tobacco due to a lack of such data that impacts the BMI of an individual. Another limitation is the absence of alcohol consumption as a covariate in the analysis. Given its frequent co-occurrence with tobacco use and its independent effect on nutritional outcomes, the inability to control for alcohol may have introduced residual confounding in the observed association between tobacco use and BMI. One limitation of this study is that the sample composition is predominantly female (87.8%) with a smaller proportion of males (12.2%), which deviates from the gender distribution in the general population. This gender imbalance may impact the generalizability of our findings to broader populations, particularly male subgroups. Future research should prioritize balanced gender representation to enhance generalizability. We can also not comment on other factors, such as weight cycling, that could also be involved in our study population.

There are certain policy recommendations emerging from the study. The present study underscores the urgent need for integrated interventions addressing tobacco use and weight-related health issues. A multi-pronged approach is essential to translate these findings into public health improvements. Firstly, comprehensive nutrition and weight management support must be integrated into smoking cessation programs. This includes developing standardised protocols within cessation clinics that offer nutritional counselling, personalized dietary plans, and behavioural strategies for weight management. Cessation counselors should be trained to address weight-related concerns, providing practical tools for maintaining a healthy weight during and after quitting. Mobile health applications and digital platforms can further enhance support and track progress in both smoking abstinence and weight management. Secondly, smoking cessation interventions within Tuberculosis (TB) clinics should be strengthened. Routine tobacco use screening should be implemented for all TB patients, with smoking cessation counselling integrated into standard TB treatment protocols. Targeted nutritional support is crucial in recognising the impact of smoking on BMI and treatment outcomes. Collaborative care models linking TB clinics with nutrition and weight management services can optimise patient care. Furthermore, targeted public health campaigns are needed to emphasise the combined negative impacts of tobacco use and malnutrition, especially in vulnerable communities. These campaigns should utilise culturally relevant messaging and diverse communication channels to reach diverse populations, including younger generations, and should incorporate visual aids and testimonials highlighting the benefits of quitting smoking and maintaining a healthy weight. Partnering with community leaders and organisations will aid in disseminating information and promoting healthy behaviours, including encouraging exercise as part of smoking cessation programs to alleviate withdrawal symptoms and prevent weight gain. Besides, future studies with more detailed behavioural and dietary data should aim to account for co-occurring risk factors like alcohol consumption to further clarify the relationship between tobacco use and BMI. Concurrently, healthcare guidelines and clinical practice should be updated at national and regional levels to include standardised screening tools for tobacco habits and BMI status for all outpatient department (OPD) patients, enabling the identification and management of at-risk individuals. Healthcare professionals should receive training on assessing and managing tobacco use and weight-related issues, and electronic health record (EHR) systems should be implemented to facilitate tracking and monitoring of both smoking status and BMI. Finally, it's crucial to prioritize obesity prevention strategies in younger generations within schools, colleges, and community centers. This involves developing educational programs that promote healthy lifestyles, including balanced diets and regular physical activity, and utilizing social media and digital platforms to engage young people in healthy lifestyle promotion. By implementing these recommendations, we can effectively address the complex interplay between tobacco use and weight-related health challenges, leading to improved health outcomes and a healthier population.

**To conclude,** India, being among the largest tobacco consumers in the world, is also witnessing changes in obesity prevalence, which raises concerns regarding the relationship between the two. The present study is the first of its kind from India to have ever documented the potential relationship between tobacco type (both smoking and smokeless) with BMI status using a nationally representative large sample size. Our Adjusted regression analysis results mainly point towards the BMI-lowering effect of tobacco, which is more pronounced for smokeless forms, and the PSM analysis further confirmed these findings. At the same time, the proportion of overweight and obese participants was highest in the smokers, highlighting the role of other concurrent risk factors. These results highlight the nutritional vulnerability of the tobacco users and underscores the need for targeted nutritional interventions among tobacco users to address the dual burden of tobacco use and undernutrition. It is high time to raise national concern and implement targeted policy guidelines aimed at reducing harmful tobacco consumption. At the same time, there is an urgent need to increase awareness about healthy body weight across all age groups to effectively move toward achieving our national goals for non-communicable disease (NCD) prevention and control.

## Supporting information

**S1 Table. Multinomial binary logistic regression analysis depicting the likelihood of being underweight or over-weight/obese in participants with or without tobacco use (any form).**
(DOCX)

## Author contributions

**Conceptualization:** Madhur Verma, Nitin Kapoor, Pritam Halder, Rakesh Kakkar, Sanjay Kalra.

**Data curation:** Madhur Verma.

**Formal analysis:** Ajit Kumar Jaiswal, Pritam Halder.

**Investigation:** Ajit Kumar Jaiswal, Vandana Esht, Waseem Mumtaz Ahamed, Sonu Goel.

**Methodology:** Madhur Verma, Nitin Kapoor, Ajit Kumar Jaiswal.

**Resources:** Ajit Kumar Jaiswal, Prakash Kumar.

**Software:** Ajit Kumar Jaiswal, Prakash Kumar.

**Supervision:** Rakesh Kakkar, Sanjay Kalra, Sonu Goel.

**Validation:** Nitin Kapoor, Rakesh Kakkar, Sanjay Kalra, Sonu Goel.

**Visualization:** Nitin Kapoor, Rakesh Kakkar, Sanjay Kalra.

**Writing – original draft:** Madhur Verma, Nitin Kapoor, Prakash Kumar, Pritam Halder, Vandana Esht, Waseem Mumtaz Ahamed, Omna Singh.

**Writing – review & editing:** Madhur Verma, Nitin Kapoor, Pritam Halder, Rakesh Kakkar, Sanjay Kalra, Sonu Goel.

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
