## [Decision Letter · Decision Letter 0]

11 Feb 2025

PONE-D-24-52387Relationship Between Tobacco Use and Body Mass Index Among Indian Adults- Findings of National Family and Household Survey (NFHS-5)PLOS ONE

Dear Dr. Goel,

Thank you for submitting your manuscript to PLOS ONE. After careful consideration, we feel that it has merit but does not fully meet PLOS ONE’s publication criteria as it currently stands. Therefore, we invite you to submit a revised version of the manuscript that addresses the points raised during the review process.

Dear Authors,

The reviewers have a mixed response to the manuscript. You are requested to kindly go through the suggestions, especially reviewer number 3 and submit the revised version for further consideration.

We look forward to receiving your revised manuscript.

Kind regards,

Yogesh Kumar Jain, MPH

Academic Editor

PLOS ONE

Reviewers' comments:

Reviewer's Responses to Questions

**Comments to the Author**

1. Is the manuscript technically sound, and do the data support the conclusions?

Reviewer #1: Yes

Reviewer #2: Yes

Reviewer #3: No

2. Has the statistical analysis been performed appropriately and rigorously? 

Reviewer #1: Yes

Reviewer #2: Yes

Reviewer #3: No

3. Have the authors made all data underlying the findings in their manuscript fully available?

Reviewer #1: No

Reviewer #2: No

Reviewer #3: Yes

4. Is the manuscript presented in an intelligible fashion and written in standard English?

Reviewer #1: Yes

Reviewer #2: Yes

Reviewer #3: Yes

5. Review Comments to the Author

Reviewer #1: This is my review for the manuscriopt"Relationship Between Tobacco Use and Body Mass Index Among Indian Adults- Findings of National Family and Household Survey (NFHS-5)"

Dear authors,thank you for your effort, it's a very good research

- key words: Body mass index must be add.

-In table 1 and 3: regarding the categories of obesity, I think that underweight will be better than undernourished

-I think the recommendation section need to be more comperhensive and directed to the practical implication of the findings.

Reviewer #2: The study has various strengths with the large sample size and using robust and standardized data collection methodology and analysis, which makes it scientifically sound to draw the conclusions. However, one concern I have is that the study sample is 87.8% female and 12.2% male. This skew in the gender proportion from the general population can affect the generalizability of the study findings. As it is not possible to amend this at this stage, it should be mentioned in the limitation.

Second, although the main finding and conclusion of the paper highlights the BMI-lowering effect of tobacco, the conclusion should also address (both in the main paper and in the abstract), the finding of the high prevalence of obesity/overweight in the smoker group with some further analysis.

Reviewer #3: General Comments:

The authors have undertaken an important analysis using NFHS-5 data to explore the association between tobacco use and BMI. While the study presents valuable insights, there are several areas where the manuscript can be improved to enhance the clarity, robustness, and validity of the findings.

Specific Comments:

1) The manuscript would benefit from a clear flow diagram outlining the sample selection process, particularly in categorizing tobacco users by type of tobacco used.

2) The official NFHS-5 report (Chapter 11, Page 445) states that 39% of men and 4% of women aged 15-49 use any form of tobacco. However, the authors report a tobacco use prevalence of approximately 7%. This discrepancy needs to be clarified, as it raises concerns about data accuracy and representativeness.

3) The comparison should be made primarily among men due to the more balanced distribution of tobacco users and non-users in this group.

4) The NFHS-5 report explicitly states that tobacco use decreases with increasing wealth quintiles. Specifically, 22% of men in the highest wealth quintile use tobacco, compared to 59% of men in the lowest quintile. Additionally, 9% of women in the lowest wealth quintile use tobacco. These socioeconomic patterns must be considered when interpreting the relationship between tobacco use and undernutrition.

5) A more robust statistical approach is required to control for potential confounders that influence BMI independently of tobacco use.

6) Given the large NFHS-5 sample size, Propensity Score Matching (PSM) should be considered, matching participants by residence (urban/rural), income quintile, age group, and educational level to strengthen causal inference.

7) Alcohol consumption should be included as a covariate, as tobacco use and alcohol consumption are often correlated. Controlling for alcohol use will help isolate the effect of tobacco on BMI.

8) The approach of first classifying individuals into underweight, normal, and overweight categories and then examining tobacco use patterns suggests potential reverse causality or selection bias. Instead, the authors should compare BMI between tobacco users and non-users before categorizing BMI.

9) In Table 2, the analysis within tobacco users categorized as normal, underweight, and overweight is less informative. A more appropriate comparison would be between tobacco users and non-users, as tobacco use itself can act as a confounder.

10) The authors should provide regression model results with and without tobacco use as an independent variable. A supplementary table should illustrate how the inclusion of tobacco use affects BMI estimates, allowing readers to assess its actual contribution.

6. PLOS authors have the option to publish the peer review history of their article (what does this mean? ). If published, this will include your full peer review and any attached files.

**Do you want your identity to be public for this peer review?** For information about this choice, including consent withdrawal, please see our Privacy Policy .

Reviewer #1: No

Reviewer #2: No

Reviewer #3: **Yes: ** Nebyu Amaha

---

## [Author Response · Author response to Decision Letter 1]

31 Mar 2025

Comment 1. 2. We note that you have indicated that there are restrictions to data sharing for this study. PLOS only allows data to be available upon request if there are legal or ethical restrictions on sharing data publicly. For more information on unacceptable data access restrictions, please see http://journals.plos.org/plosone/s/data-availability#loc-unacceptable-data-access-restrictions.

Response: We used publicly available datasets that are available through a proper procedure and have been mentioned in the updated data-sharing statement

Comment 2. 3. Your ethics statement should only appear in the Methods section of your manuscript. If your ethics statement is written in any section besides the Methods, please move it to the Methods section and delete it from any other section. Please ensure that your ethics statement is included in your manuscript, as the ethics statement entered into the online submission form will not be published alongside your manuscript.

Response: We have now complied with the instructions.

Comment 3. 5. Review Comments to the Author: Please use the space provided to explain your answers to the questions above. You may also include additional comments for the author, including concerns about dual publication, research ethics, or publication ethics. (Please upload your review as an attachment if it exceeds 20,000 characters)

Response: We have now complied with the instructions.

Reviewers' comments:

Reviewer's Responses to Questions

Comments to the Author

Reviewer #1:

Comment 4. This is my review for the manuscript "Relationship Between Tobacco Use and Body Mass Index Among Indian Adults- Findings of National Family and Household Survey (NFHS-5)" Dear authors, thank you for your effort, it's a very good research.

Response: We are thankful for taking an interest in the study.

Comment 5. Keywords: Body mass index must be added.

Response: We have now added it in the keywords section below the abstract.

Comment 6. -In Table 1 and 3: regarding the categories of obesity, I think that underweight will be better than undernourished

Response: We are thankful for your observations. We have now made suggested changes throughout the manuscript.

Comment 7. I think the recommendation section needs to be more comprehensive and directed to the practical implications of the findings.

Response: We have now updated our recommendations extensively based on the revised analysis.

Reviewer #2:

Comment 8. The study has various strengths with the large sample size and using robust and standardised data collection methodology and analysis, making it scientifically sound to draw conclusions. However, one concern I have is that the study sample is 87.8% female and 12.2% male. This skew in the gender proportion from the general population can affect the generalizability of the study findings. As it is not possible to amend this at this stage, it should be mentioned in the limitation.

Response: the survey has a disproportionate sample size with a higher number of females than males, as the survey was originally designed to collect robust data that was important for maternal and child health-related policy decision-making. Over the period of time, as the priorities evolved, men were included in the survey to have a broader perspective of maternal and child health, and later, the number of variables also increased allowed us to do the present study. We have now added this to our limitations as per your suggestions.

Comment 9. Second, although the main finding and conclusion of the paper highlights the BMI-lowering effect of tobacco, the conclusion should also address (both in the main paper and in the abstract), the finding of the high prevalence of obesity/overweight in the smoker group with some further analysis.

Response: We agree to your comment. We have now added this to our conclusion in main text. And this is mentioned in the conclusions of the abstract section as well.

Reviewer #3:

Comment 10. General Comments: The authors have undertaken an important analysis using NFHS-5 data to explore the association between tobacco use and BMI. While the study presents valuable insights, there are several areas where the manuscript can be improved to enhance the clarity, robustness, and validity of the findings.

Response: Thank you for taking an interest in the study. Your comments were really helpful and helped us to improve our manuscript significantly.

Specific Comments:

Comment 11. The manuscript would benefit from a clear flow diagram outlining the sample selection process, particularly in categorising tobacco users by type of tobacco used.

Response: We have now added figure 1 as per your suggestions.

Comment 12. The official NFHS-5 report (Chapter 11, Page 445) states that 39% of men and 4% of women aged 15-49 use any form of tobacco. However, the authors report a tobacco use prevalence of approximately 7%. This discrepancy must be clarified, as it raises concerns about data accuracy and representativeness.

Response: Thank you for raising this comment. While the report included the whole sample, we only included those with complete BMI (height and weight data) and tobacco usage data and also excluded pregnant females, which led to the changes in our estimates. We have further redone the analysis as per your suggestions and confirmed the concern raised by you.

Comment 13. The comparison should be made primarily among men due to the more balanced distribution of tobacco users and non-users in this group.

Response: We acknowledge the reviewer’s suggestion to conduct a separate analysis among men due to the more balanced distribution of tobacco users and non-users in that group. While tobacco use is more prevalent in men, its impact on women is more severe. This has been added to the text in the discussion section. However, our primary aim was to assess the association between tobacco use and nutritional outcomes in the general adult population. While we recognise that tobacco use is more prevalent among men, our multivariable models accounted for sex and other potential confounders, such as education, wealth, and region. Moreover, stratifying the analysis by sex would have substantially reduced statistical power in the female subgroup and diverted focus from our broader population-level perspective. Nonetheless, we have now highlighted the sex disparity in tobacco use in the limitations section and advised caution in interpreting pooled estimates, particularly for subgroups with low prevalence of tobacco use.

Comment 14. The NFHS-5 report explicitly states that tobacco use decreases with increasing wealth quintiles. Specifically, 22% of men in the highest wealth quintile use tobacco, compared to 59% of men in the lowest quintile. Additionally, 9% of women in the lowest wealth quintile use tobacco. These socioeconomic patterns must be considered when interpreting the relationship between tobacco use and undernutrition.

Response: Since tobacco use is more prevalent in lower wealth quintiles—where undernutrition is also more common—the wealth status was adjusted for in all statistical models to minimize confounding.

Comment 15. A more robust statistical approach is required to control for potential confounders that influence BMI independently of tobacco use. Given the large NFHS-5 sample size, Propensity Score Matching (PSM) should be considered, matching participants by residence (urban/rural), income quintile, age group, and educational level to strengthen causal inference.

Response: We thank you for your suggestion. To minimize confounding and improve causal inference, we implemented propensity score matching using key sociodemographic characteristics like age group, education level, place of residence, and wealth quintile and the emerging results are insightful.

Comment 16. Alcohol consumption should be included as a covariate, as tobacco use and alcohol consumption are often correlated. Controlling for alcohol use will help isolate the effect of tobacco on BMI.

Response: We acknowledge that alcohol consumption is often correlated with tobacco use and may act as a confounder in the tobacco–BMI relationship. Unfortunately, detailed and reliable alcohol consumption data were not available for the full sample in NFHS-5, and underreporting, particularly among women, is a known limitation. While we could not control for alcohol use directly, we recognise this as an important limitation and have included it in the revised manuscript (see Limitations section lines 15-17 of the paragraph).” Further, we’ve added some future scope of work considering alcohol in lines 402-403.

Comment 17. The approach of first classifying individuals into underweight, normal, and overweight categories and then examining tobacco use patterns suggests potential reverse causality or selection bias. Instead, the authors should compare BMI between tobacco users and non-users before categorising BMI.

Response: To address reverse causality concerns, we conducted an additional analysis comparing mean BMI between tobacco users and non-users prior to categorisation. We conducted an independent t-test to acknowledge the above comment. The results have been added to the result section text just before the description of Table 3.

Comment 18. In Table 2, the analysis of tobacco users categorised as normal, underweight, and overweight is less informative. A more appropriate comparison would be between tobacco users and non-users, as tobacco use itself can act as a confounder.

Response: We appreciate the reviewer’s suggestion. As our paper revolves around obesity due to tobacco use, we first wanted to describe the obesity-related characteristics in our manuscript. So, while Table 2 shows BMI distribution across background characteristics, Table 3 presents a distribution of participants as per their BMI category and stratified by different socio-demographic variables across different types of tobacco users (smoked, smokeless, dual) and includes both users and non-users. It allows for comparison between tobacco users and non-users in relation to underweight and overweight/obesity status across various sociodemographic groups. Therefore, we believe Table 3 sufficiently addresses this concern and enables a more nuanced interpretation of the tobacco–BMI relationship. Please suggest if it still needs modifications.

Comment 19. The authors should provide regression model results with and without tobacco use as an independent variable. A supplementary table should illustrate how the inclusion of tobacco use affects BMI estimates, allowing readers to assess its actual contribution.

Response: Thank you for raising this concern. We have now done another regression analysis with tobacco use in any form (yes/no) to assess the likelihood of being underweight or overweight/obese in the supplementary file (i.e., supplementary table 1).- “Both unadjusted and adjusted binary logistic regression analyses reveal a significant association between tobacco use and nutritional status. In the unadjusted model, tobacco users had 15% and 8% lower odds of being underweight (OR: 0.85; 95% CI: 0.83–0.87) and overweight (0.92; 0.91–0.90) than non-users. However, after adjusting for sociodemographic factors, the association reversed: tobacco users had 20% higher odds of being underweight (AOR: 1.2; 95% CI: 1.2–1.2) and 10% lower odds of being overweight (0.9; 0.9–0.9).”

---

## [Decision Letter · Decision Letter 1]

6 Apr 2025

Relationship Between Tobacco Use and Body Mass Index- A Propensity Score Matching Analysis of an Indian National Survey

PONE-D-24-52387R1

Dear Dr. Goel,

We’re pleased to inform you that your manuscript has been judged scientifically suitable for publication and will be formally accepted for publication once it meets all outstanding technical requirements.

Kind regards,

Yogesh Kumar Jain, MPH

Academic Editor

PLOS ONE

Additional Editor Comments (optional):

Reviewers' comments:

Reviewer's Responses to Questions

**Comments to the Author**

1. If the authors have adequately addressed your comments raised in a previous round of review and you feel that this manuscript is now acceptable for publication, you may indicate that here to bypass the “Comments to the Author” section, enter your conflict of interest statement in the “Confidential to Editor” section, and submit your "Accept" recommendation.

Reviewer #1: All comments have been addressed

Reviewer #2: All comments have been addressed

2. Is the manuscript technically sound, and do the data support the conclusions?

Reviewer #1: Yes

Reviewer #2: Yes

3. Has the statistical analysis been performed appropriately and rigorously? 

Reviewer #1: Yes

Reviewer #2: I Don't Know

4. Have the authors made all data underlying the findings in their manuscript fully available?

Reviewer #1: No

Reviewer #2: Yes

5. Is the manuscript presented in an intelligible fashion and written in standard English?

Reviewer #1: Yes

Reviewer #2: Yes

6. Review Comments to the Author

Reviewer #1: Dear authors; Thank you for addressing all comments. Wishing you all luck in this research and all future work

Reviewer #2: I confirm the author has incorporated both comments I have provided. They have indicated the skewed gender distribution as a limitation. They have also highlighted the additional finding of the high prevalence of obesity/overweight in the smoker group in addition to the main conclusion of the BMI-lowering effect of tobacco.

7. PLOS authors have the option to publish the peer review history of their article (what does this mean? ). If published, this will include your full peer review and any attached files.

**Do you want your identity to be public for this peer review?** For information about this choice, including consent withdrawal, please see our Privacy Policy .

Reviewer #1: No

Reviewer #2: **Yes: ** Lia Tadesse Gebremedhin

---

## [Editor Report · Acceptance letter]

PONE-D-24-52387R1

PLOS ONE

Dear Dr. Goel,

I'm pleased to inform you that your manuscript has been deemed suitable for publication in PLOS ONE. Congratulations! Your manuscript is now being handed over to our production team.

Kind regards,

on behalf of

Dr. PLOS Manuscript Reassignment

Staff Editor

PLOS ONE